# The Mediating Role of Contextual Problems and Sensation Seeking in the Association between Substance Use and Mental Health in Adolescents from Northern Chile

**DOI:** 10.3390/ijerph19042262

**Published:** 2022-02-17

**Authors:** Alejandra Caqueo-Urízar, Diego Atencio-Quevedo, Alfonso Urzúa, Jerome Flores, Matías Irarrázaval

**Affiliations:** 1Instituto de Alta Investigación, Universidad de Tarapacá, Arica 1000000, Chile; 2Escuela de Psicología y Filosofía, Universidad de Tarapacá, Arica 1000000, Chile; datencioq@gmail.com (D.A.-Q.); jflores@uta.cl (J.F.); 3Centro de Justicia Educacional (CJE), Pontificia Universidad Católica de Chile, Santiago 7820436, Chile; 4Escuela de Psicología, Universidad Católica del Norte, Antofagasta 1270709, Chile; alurzua@ucn.cl; 5Departamento de Psiquiatría, Facultad de Medicina, Hospital Clínico, Universidad de Chile, Santiago 8380453, Chile; mirarrazavald@u.uchile.cl; 6Institute for Depression and Personality Research MIDAP, Santiago 8380453, Chile

**Keywords:** substance use, mental health, contextual problems, sensation seeking, youth, adolescents

## Abstract

Substance use is a risk behavior that has been associated with adverse mental health outcomes in adolescence. The aim of this study was to determine the relation between behavioral problems, emotional problems, and substance use as well as the mediating role of contextual problems and sensation seeking in this relation. A cross-sectional study of 2277 adolescents from Northern Chile was conducted. The System for the Evaluation of Children and Adolescents (SENA) was used to assess substance use, contextual problems, sensation seeking, and emotional and behavioral problems. Through a mediational model, it was observed that substance use has a positive indirect effect on emotional and behavioral problems when both contextual problems and sensation seeking act as mediating variables. An indirect effect of substance use on contextual problems with sensation seeking as a mediator was also observed. The results suggests that context and sensation seeking are a relevant source of information in understanding adolescents and their propensity to use drugs. Interventions based on addressing contextual problems (problems with school, peers, and family) and enhancing personal resources should be implemented in order to reduce substance use in adolescents as well as the consequences it can generate in the short, medium, and long term.

## 1. Introduction

Adolescence is a phase characterized by major biological, emotional, cognitive, and social changes [1]. This transition period brings with it risk factors that begin to manifest themselves, which can lead to the appearance of problems that can have severe consequences for the rest of life [2], such as psychological disorders and problematic patterns of substance use [3]. As a result, drug use in adolescence has become a public health problem, since the consequences on physical, psychological, and social well-being are highly disabling in the short and long term [4]. 

Evidence in the literature has shown that chronic exposure to alcohol and drug use is associated with neural changes in the brain’s reward circuitry, independent of the lifecycle stage of the user [5]. However, due to the neuromaturation-related processes of adolescence, an early onset of alcohol and drugs consumption makes young people more susceptible to severe presentations of neurodegenerative processes and neuropsychological deficits, even when consumption patterns do not meet the criteria to be categorized as problematic [6,7].

Beyond the changes in neuroanatomy that substance use in adolescence can cause, early onset (before the age of 14) has been associated with a negative impact at the psychosocial level, either at the emotional level, with an increased risk of depression, anxiety, social phobia, and suicidal ideation, as well as increased rates of school dropout and delinquent and antisocial behaviors [8]. In the same sense, a recent study carried out by Sánchez-García et al. showed an association between the use of tobacco and alcohol with the presence of schizotypal traits, which is a prelude and risk factor for the appearance of psychotic spectrum disorders [9].

Regarding the emotional area, several studies indicate the existence of a positive association between anxious and depressive symptoms and early consumption of different substances, such as tobacco [10,11,12,13,14], marijuana [13,15], and alcohol [14,15,16,17,18]. On the other hand, opposite associations were found in a study conducted with adolescents in New England, where it was observed that the presence of anxious and depressive symptoms was associated with a significantly higher risk of substance use [16]. A systematic review conducted by Lemyre et al. [19] showed that in the case of social anxiety, in addition to being associated with increased tobacco and marijuana use, there was also evidence suggesting a relationship between social anxiety and the use of prescription drugs as a source of coping with symptoms.

Concerning behavioral problems, the literature reports a bidirectional relationship between these and drug use. On one hand, a meta-analytic review found evidence that children and adolescents diagnosed with Attention Deficit Hyperactivity Disorder (ADHD) are more likely to present use disorders and dependence on nicotine, alcohol, marijuana, cocaine, and other substances [20]. On the other hand, there is evidence that substance use in adolescence has been associated with aggressive behavior, impulse control problems, and antisocial behavior [21,22,23].

Recently, adolescent substance use has been investigated as a complex phenomenon, in which individual and contextual variables are constituted as risk factors, acting as mediators in the mental health effects of substance use [24].

At an individual level, sensation seeking, which can be understood as a need to experience varied and complex sensations, coupled with a desire to take physical and social risks in order to enjoy such experiences, has been linked to substance use in adolescence [25]. Sensation seeking has been associated with addictive behaviors, such as problematic alcohol and drug use [26,27,28], as well as with Internet addictions [29]. On the other hand, this variable has also been related to mental health variables, both to emotional problems, such as higher levels of depression, anxiety, post-traumatic stress disorder, and suicidal risk [30]; and to behavioral problems, mainly antisocial behaviors, impulsivity, aggressiveness, and anger management problems [28,31,32].

The impact of social and situational background of adolescents on substance use has been an incipient but growing field of research [33,34,35,36]. In this regard, it has been found that problems with peers have been related to an increase in substance use, because social pressure and the search for a sense of belonging make adolescents more likely to consume substances [36,37,38]. In addition, variables related to the family context have been associated with problematic substance use patterns, either due to the presence of coexisting problems, such as child–parent violence [39,40], or due to the lack of supervision and involvement of caregivers [36,41].

In Chile, alcohol and drug use in adolescents is an issue that has been periodically documented nationally by the Ministry of Education, the National Council for Drug Control (CONACE), and the National Service for the Prevention and Rehabilitation of Drug and Alcohol Use (Servicio Nacional para la Prevención y Rehabilitación del Consumo de Drogas y Alcohol, SENDA) since 1995 [42].

The Thirteenth National Study on Drugs in the Chilean School Population 2019 [42] reported a concerning reality in the progress of the consumption patterns of the country’s adolescent population. Although alcohol and tobacco consumption decreased, the frequency of binge drinking (consuming more than five alcoholic drinks in an occasion) increased significantly. Similarly, the study reported an increase in the use of cocaine base paste, ecstasy, hallucinogens, and prescription tranquilizers and stimulants, which was statistically significant compared to the previous measurement [42].

Among the contextual factors associated with these consumption patterns, it was observed that access to these substances was given by peers at parties, in their homes or in and around educational establishments [42]. Another factor of major relevance is that parents’ perception of alcohol and marijuana use was less negative than in previous measurements, generating instances of less supervision and, therefore, a risk factor for the use of these substances [42].

Although the magnitude of this study makes it a valuable source of information for understanding the consumption patterns of adolescents in the country, it does not report the impact of these patterns on mental health indicators, emotional, and behavioral problems in this population. Moreover, no causal relationships are established between the contextual factors reported and the consumption of alcohol and drugs in the sample studied.

The educational context is the one where adolescents spend a considerable amount of time, so that educational establishments become an intermediary between governmental institutions and students, where, taking an ecological model as a frame of reference, the school context becomes the most appropriate place for education and socialization in terms of adolescent substance use [43]. Therefore, the purpose of this study is to determine the mediating role that sensation seeking and contextual variables have in the relationship between substance use and indicators of emotional and behavioral problems in adolescents from northern Chile.

**Hypothesis** **1** **(H1).**
*Substance use has a direct effect on sensation seeking and contextual problems.*


**Hypothesis** **2** **(H2).**
*Substance use has a direct effect on emotional and behavior problems.*


**Hypothesis** **3** **(H3).**
*Substance use has an indirect effect on emotional and behavior problems with sensation seeking and contextual problems as mediator variables.*


## 2. Materials and Methods

### 2.1. Design

A non-experimental study with a predictive cross-sectional design was conducted since all variables were measured at a single point in time, and the purpose of the study was to explore the functional relationship through the prediction of a criterion variable from one or more predictors [44].

### 2.2. Participants

The sample consisted of 2277 students from seventh grade to senior year from educational establishments in the city of Arica, Northern Chile. It was a convenience sample. Thirty-five municipal, private-subsidized and private-paid educational establishments in Arica were invited to participate in this study by the researchers’ team. Finally, 29 schools of the city participated in the study and 25 were considered for this paper, as four educational establishments only had elementary school. Informed consent was obtained from parents and from the adolescents. Ages ranged from 12 to 18 years, with a mean of 14.4 years (SD1.7). Grade distribution was as follows: 20.7% (*n* = 471) were from 7th grade, 20.2% (*n* = 460) were from 8th grade, 17.9% (*n* = 407) were from 9th grade, 15.7% (*n* = 358) were from 10th grade, 14.2% (*n* = 324) were from 11th grade and, 11.3% (*n* = 257) were from 12th grade. Regarding the gender of the students, 50.4% (*n* = 1148) were girls and 49.6% (*n* = 1129) were boys. Most of the educational establishments belonged to a population of medium and/or low socioeconomic level, with 17 schools belonging to a high socioeconomical vulnerability and 8 schools belonging to low vulnerability. Finally, 93.3% (*n* = 2124) of the students were Chilean, followed by 3.1% (*n* = 70) from Bolivia, 2.3% (*n* = 53) from Perú, and 1.4% (*n* = 30) from other countries of the region. No prior mental health records were considered in this study.

### 2.3. Instruments

The ad hoc sociodemographic scale included gender, age, and grade of the participants (see Appendix A).

Child and Adolescent Assessment System (Sistema de Evaluación de Niños y Adolescentes; SENA) [45]: It is an instrument developed by specialists in psychopathology and psychological assessment whose purpose is to measure a wide range of emotional and behavioral problems in people aged 3 to 18 years. The items have a 5-level Likert scale response format, ranging from 1 (never or almost never) to 5 (always or almost always). The following scales of the self-report version for high school (12–18 years old) were used in this study: (a) Emotional Problems, composed of the subscales that assess depression, anxiety, social anxiety, somatic symptoms, post-traumatic stress, and obsessive-compulsive symptoms; (b) Behavioral Problems, that assess the presence of attention problems, hyperactivity–impulsivity, anger control problems, aggression, defiant behavior, and antisocial behavior; (c) Contextual Problems, evaluating problems with family, school, and peers; (d) Sensation Seeking, which assesses the tendency of individual to crave novel experiences, which may be potentially dangerous; € Substance Use, which aims to detect the use of alcohol, marijuana, and other drugs in adolescents. Higher scores indicate that participants have more difficulties in these areas. Recently, Sanchez-Sanchez et al. [46] have found that the reliability of its subscales is above 0.7 in Spain.

### 2.4. Procedures

This research was approved by the Scientific Ethics Committee of the Universidad de Tarapacá (26-2017).

Principals and counselors of 35 educational establishments in Arica were contacted and invited to participate in this study voluntarily by the investigator’s team. Municipal, private-subsidized, and private-paid schools were included. Finally, in the second semester of 2018, students from 29 schools of the city participated in the study. Informed consent was obtained from parents. Subsequently, consent was also obtained from the adolescents. Then, the instruments were applied after scheduling the dates of application in a group setting in the classroom. At least two trained surveyors were present to answer any questions, together with a teacher from the same course. The duration of the evaluation was approximately 45 min. Among those authorized, 97% agree to answer the survey.

### 2.5. Data Analysis

Initially, to process the missing values, the missing value was replaced with the mean of the scale in those instruments with less than 3% missing data. In the case of the possible outliers, a median imputation strategy was used, replacing extremes values with the median value of the scale. As exclusion criteria, instruments with more than 3% of missing data were removed from the base, leaving a definitive sample of 2277 students.

The missing value pattern was examined and found to be completely random (MCAR). Consequently, the strategy of replacing the values by the mean is acceptable [47].

To characterize the sample, descriptive analyses were performed on categorical variables (number and percentage) and quantitative variables (means and standard deviations). Subsequently, the relationships between the study variables were evaluated by means of a Pearson correlation matrix.

A path analysis was carried out to determine the direct and indirect effects between the studied variables. In the assessment of the model, the classical criteria for the interpretation of model fit were taken as a reference, considering a Root Mean Square Error of Approximation (RMSEA) less than 0.08 as acceptable and less than 0.06 as good, a Standardized Root Mean Square (SRMR) less than 0.8 as adequate, a Comparative Fit Index (CFI) and a Tucker–Lewis Index (TLI) greater than 0.9 as adequate, and above 0.95 as optimal [48,49,50]. Since chi-square is not currently considered an indispensable indicator given the problems it presents with samples larger than 200 and in cases of non-normal distribution [48,49], it was not considered, although it is reported.

Statistical analyses were carried out with version 23 of the statistical package IBM SPSS Statistics [51] and with IBM SPSS and with AMOS Graphic, version 26 [52]. Figure 1 shows a graphical representation of the proposed model to explain the association between the variables.

## 3. Results

Table 1 shows the descriptive statistics of the studied variables and differences in means according to vulnerability level of the school. Regarding these differences, only sensation seeking presented a significant difference, where high-vulnerability students presented a higher score on this variable. It can be observed that the levels of skewness and kurtosis are not within the expected ranges, indicating that the variables are not normally distributed [53]. In large samples, normality is inessential, since Central Limit Theorem assures that the sampling distribution of the estimates will converge toward a normal distribution as the sample size increases [54]. However, to increase the accuracy of predictions and add confidence intervals, a bootstrap was performed, with 5000 bootstrap samples and a 95% bias-corrected confidence level.

To estimate the correlations between emotional problems, behavior problems, contextual problems, sensation seeking, and substance use, a Pearson correlation matrix is presented (Table 2). It can be observed that emotional problems have a large direct effect on contextual problems (r > 0.5) [55], a medium direct effect on behavior problems and sensation seeking (r > 0.3) [55], and a small direct effect on substance use (r < 0.3) [55]. In the case of contextual problems, it presents a large direct effect on contextual problems and sensation seeking (r > 0.5) [55] and a medium direct effect on substance use (r > 0.3) [55]. Contextual problems presented a medium direct effect on sensation seeking (r > 0.3) [52], and a small direct effect on substance use (r < 0.3) [55]. Finally, sensation seeking had a medium direct effect on substance use (r < 0.3) [55].

Since all variables presented significant effects, a mediation path analysis was developed, using substance use as a predictor, sensation seeking and contextual problems as mediators, and emotional and behavior problems as dependent variables.

Regarding direct effects, as seen in Table 3, substance use has a direct effect on sensation seeking (β = 0.396, *p* = 0.003), contextual problems (β = 0.132, *p* = 0.003), behavior problems (β = 0.156, *p* = 0.003), and emotional problems (β = −0.059, *p* = 0.011). Sensation seeking has a direct effect in contextual problems (β = 0.300, *p* = 0.006), behavior problems (β = 0.289, *p* = 0.007), and emotional problems (β = 0.141, *p* = 0.008). On the other hand, contextual problems have a direct effect on behavior problems (β = 0.495, *p* = 0.005) and emotional problems (β = 0.562, *p* = 0.004).

Table 4 shows standardized indirect effects. Substance use has an indirect effect on contextual problems (β = 0.119, *p* = 0.004), behavior problems (β = 0.238, *p* = 0.004), and on emotional problems (β = 0.197, *p* = 0.005). Finally, sensation seeking has an indirect effect on behavior problems (β = 0.148, *p* = 0.003) and emotional problems (β = 0.168, *p* = 0.006). Figure 2 shows a graphical representation of the obtained model to explain the association between the variables.

The goodness-of-fit indicators were χ2 = 27.530 (df = 1, χ2/df = 27.530), CFI = 0.992, TLI = 0.924, and SRMR = 0.0154, which indicate a good fit of the model. RMSEA = 0.108 (90% CI: 0.076–0.045) was higher than the expected values, but since the model has a small *df*, it is usual that the RMSEA falsely indicates a poor fitting model [56], so it is recommended not to compute the RMSEA in a model with these characteristics.

## 4. Discussion

The aim of this study was to determine the mediating role that sensation seeking and contextual variables have in the association between substance use and indicators of emotional and behavioral problems in adolescents from Northern Chile.

The results suggest the existence of direct effects of substance use on sensation seeking, which seems to hint at a bidirectional relationship between both variables, since in addition to the relationship observed in previous studies, in which sensation seeking is a risk factor that predisposes adolescents to substance use [26,27,28,29], in this study, the results seem to indicate that those who use drugs would tend to see this need to experiment new experiences increase, seeking stronger and lasting effects in stronger drugs, which seems plausible taking into account the results of the SENDA study [42], where an increase in the consumption of hallucinogens, ecstasy, and stimulants and tranquilizers without medical prescription was observed in the country’s adolescents.

On the other hand, drug use had direct effects on contextual problems, which is consistent with the existing literature that indicates that drug use leads to increased episodes of violence and poor coexistence, as well as a lack of supervision and involvement among adolescents and their families [36,39,40,41]. At the same time, social pressures to fit in with the group, as well as the stigmatization of substance use and disruptive behaviors related to the effects of psychotropic drugs, lead to an increase in problems with peers and educational institutions, resulting in social isolation or expulsion from school [36,37,38].

As for the effects of substance use on emotional problems, an inverse relationship is observed, which is not consistent with previous studies that show that substance use is associated with higher prevalence of depression or anxiety [8,9,10,11,12,13,14,15,16,17,18,19]. The results of this study can be framed within the use of substances as a method of regulation and avoidance of discomfort caused by pre-existing psychological problems, which play a fundamental role both in the development of addictions and in their maintenance, producing the destructive pattern that characterizes people involved in substance misuse [57].

When the direct impact of substance use on behavioral problems is analyzed, the results are in line with the findings of other studies, confirming that the consumption of alcohol and other substances is associated with the appearance of disruptive behaviors, aggression, emotion regulation problems, inattention, and criminal and antisocial behaviors [21,22,23].

The results indicate the existence of a partial mediation of sensation seeking and contextual problems in the relationship between substance use and emotional problems, which causes the direction of the relationship between the variables to change from being negative in the direct effects to being positive when these variables are taken into account as mediators. In this way, the results become congruent with those of the existing literature, where it can be hypothesized that drugs produce a decrease in internalized symptomatology in the short term when used as a method of avoidance and emotional regulation [57]. However, in the long term, they generate a worsening of these indicators, being associated with higher levels of mood disorders and mental health problems, with the consumption of alcohol, tobacco, marijuana, and other substances being associated with the appearance of anxious–depressive symptoms [8,9,10,11,12,13,14,15,16,17,18]. In this sense, sensation seeking and contextual problems contribute to provide a better understanding of the mechanisms of substance use on the levels of depression, anxiety, and other emotional problems.

Regarding the effects of sensation seeking and contextual problems on the association between substance use and behavioral problems, it is observed that the first two have a mediating role. In the case of sensation seeking, this would generate a synergistic effect with the direct effects already caused by substance use at the behavioral level, making the appearance of aggression, anger, risky behavior, and antisocial behavior even more likely [19,20,23,26,29,30]. On the other hand, the mediating role of contextual problems is framed in the neurological effects of drug use in adolescence, generating alterations that hinder emotional regulation and make non-adaptive aggression responses more frequent in problematic situations with school, peers, or family [7,8,39,40].

The results also indicate the existence of a partial mediation of sensation seeking in the association between substance use and contextual problems; i.e., sensation seeking not only has a negative effect on individual mental health variables of adolescents who use drugs but also on their most significant environments, such as their families, schools, and peers, contributing to higher school dropout rates, social isolation, family dysfunction, and domestic violence [4,8].

The implications of the study are its contribution to the understanding of a gap in the literature at the local level, associating the findings of the SENDA study with mental health variables and the context of adolescents in the region. This can be a valuable source of information when designing interventions to work with adolescent drug use.

Among the potential interventions to be carried out, those that seek to promote protective factors and that are carried out in the school context, including teachers and parents, have been shown to be effective not only in reducing the use of tobacco, alcohol, and marijuana, as in the case of Hodder et al. [58], but are also useful because they improve variables that contribute to the well-being of adolescents and their environments, such as self-esteem and resilience [58,59].

Since the school context is where adolescents spend a considerable amount of time, the main relevance of the results of this study relates to using the educational establishment as an intermediary between government institutions and young people in the fight against problematic substance use and its adverse consequences for people’s lives; in this sense, this is in line with the study by Montgomery et al., where, taking an ecological model as a frame of reference, the school context becomes the most appropriate place for education and socialization in terms of adolescent substance use [43].

This study has limitations, among them the cross-sectional design, which does not allow us to identify or analyze trajectories of the changes and relations observed, or to establish a cause–effect explanatory model. Another limitation is that the previous mental health history of each student was not considered, since it is likely that in a sample of over 2000 participants, some of them have mental problems that are affecting the interpretation of the results. On the other hand, the data were obtained exclusively through student self-report, so studies that consider other sources of information, such as parents or teachers, are required. Future research should study the associations found longitudinally, in order to have a more complete understanding of the influence of these variables and their variations over time. In addition, young people and adolescents do not only present substance use these days, since addiction to social networks and the Internet is growing and is a new challenge [60,61,62]; therefore, new research is needed to study both substance and behavioral addictions, whether social networking or online gaming, to determine the moderating role they might present.

## 5. Conclusions

The results of this study indicate that substance use has a harmful effect not only on the emotional and behavioral indicators of adolescents but also has a significant impact on the environments in which they live. Interventions that promote the personal resources of adolescents and involve and promote conflict resolution with peers, schools, and families are key to contribute to the reduction of substance use and its long-term effects on the lives of those who start using substances at an early age.

## Figures and Tables

**Figure 1 ijerph-19-02262-f001:**
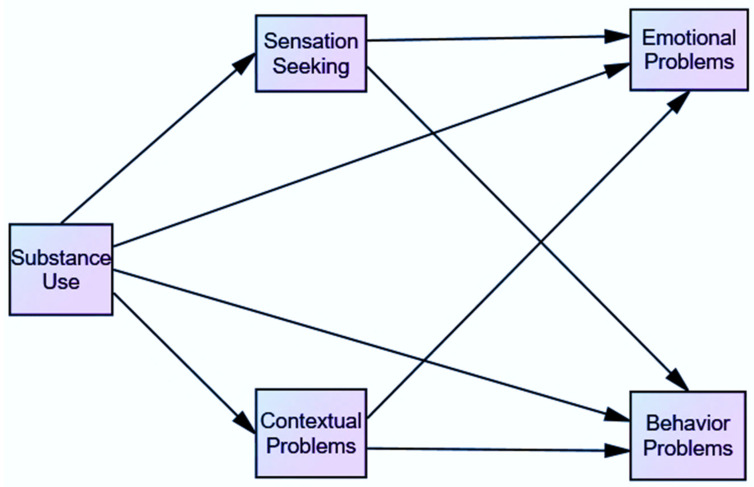
Proposed model of substance use on emotional and behavior problems.

**Figure 2 ijerph-19-02262-f002:**
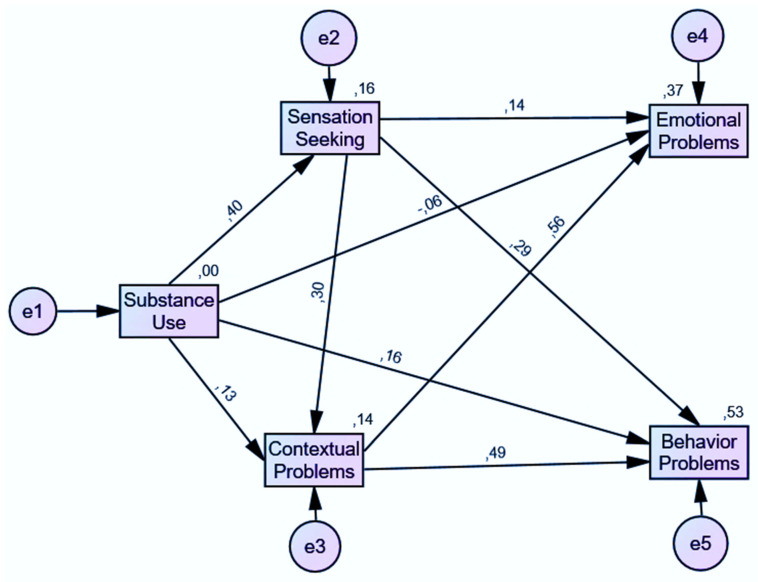
Path analysis model of substance use on emotional and behavior problems.

**Table 1 ijerph-19-02262-t001:** Studied variables descriptives and mean differences according to vulnerability level.

	Mean (SD)	Min.	Max.	Skewness	Kurtosis	Mean Difference *	*p*-Value
SUS	1.315 (0.592)	1	5	2.58	7.374	−0.013	0.594
BUS	2.299 (0.892)	1	5	0.748	−0.017	0.037	0.004
CTX	51.414 (8.623)	38.3	96.7	1.167	1.674	−0.315	0.384
CON	50.461 (9.056)	39.5	101.2	1.586	3.057	0.185	0.626
EMO	54.369 (9.832)	36.3	88.5	0.708	0.087	0.717	0.082

Note: SUS = Susbtance Use; BUS = Sensation Seeking; CTX = Contextual Problems; CON = Behavior Problems; EMO = Emotional Problems; * = Differences between low and high vulnerability schools.

**Table 2 ijerph-19-02262-t002:** Correlation matrix between studied variables.

	EMO		CON		CTX		BUS		SUS
EMO	—								
CON	0.468	***	—						
CTX	0.597	***	0.635	***	—				
BUS	0.316	***	0.525	***	0.352	***	—		
SUS	0.138	***	0.394	***	0.250	***	0.396	***	—

*** *p* < 0.001. Note: EMO = Emotional Problems; CON = Behavior Problems; CTX = Contextual Problems; BUS = Sensation Seeking; SUS = Substance Use.

**Table 3 ijerph-19-02262-t003:** Standardized direct effects.

Path	Standardized β	Lower *	Upper *	*p*-Value
SUS --> BUS	0.396	0.357	0.437	0.003
SUS --> CTX	0.132	0.084	0.180	0.003
SUS --> CON	0.156	0.118	0.203	0.003
SUS --> EMO	−0.059	−0.097	−0.019	0.011
BUS --> CTX	0.300	0.248	0.344	0.006
BUS --> CON	0.289	0.252	0.327	0.007
BUS --> EMO	0.141	0.097	0.178	0.008
CTX --> CON	0.495	0.458	0.526	0.005
CTX --> EMO	0.562	0.528	0.598	0.004

* 95% Bias-corrected confidence interval level. Note: EMO = Emotional Problems; CON = Behavioral Problems; CTX = Contextual Problems; BUS = Sensation Seeking; SUS = Substance Use.

**Table 4 ijerph-19-02262-t004:** Standardized indirect effects.

Path	Standardized β	Lower *	Upper *	*p*-Value
SUS --> CTX	0.119	0.097	0.140	0.004
SUS --> CON	0.238	0.209	0.267	0.004
SUS --> EMO	0.197	0.162	0.226	0.005
BUS --> CON	0.148	0.124	0.175	0.003
BUS --> EMO	0.168	0.138	0.196	0.006

* 95% Bias-corrected confidence interval level. Note: EMO = Emotional Problems; CON = Behavioral Problems; CTX = Contextual Problems; BUS = Sensation Seeking; SUS = Substance Use.

## Data Availability

The data presented in this study are available on request from the corresponding author.

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
