# Peer review of "The Mediating Role of Contextual Problems and Sensation Seeking in the Association between Substance Use and Mental Health in Adolescents from Northern Chile"

_ijerph, 2022, doi:10.3390/ijerph19042262_

Round 1

Reviewer 1 Report

I would like to thank the authors for an interesting article about factors affecting substance misuse in an urban environment in Northern Chile. The  observational study was limited to a single cohort over a short interval of time, and the authors properly identify this may limit the generalizability of their findings. 

The results appear to confirm what is generally accepted regarding substance use. I should note that my background is primarily related to tobacco use and measurement thereof, and my comments are primarily in this context. The findings presented provided me with a larger context in which to view tobacco use. I believe this aspect to be the most informative part of the manuscript -- seeking to understand the relationships between substance use, behavioral, and environmental factors. These aspects of the interplay between factors were most important to me as a reader. I appreciated the discussion surrounding implications for proposed interventions intended to reduce substance abuse.

While i recognize this is beyond scope of what authors proposed as the objective in their manuscript, I would be interested to learn if there are relationships between substances of abuse (e.g. drugs, alcohol, and tobacco) and if these relationships are reinforcing to one another. I realize that comment is beyond scope of the current article, but it may be an appropriate topic for a subsequent analysis, based on the description of the data collected during this study.

The statistical analysis appears properly done and the discussion points are supported by the data and analysis thereof. The conclusions appear reasonable and do not appear to be over-stated relative to the data and results presented. 

Author Response

Thank you very much for the review. The comments and changes are in the attached document. 

Reviewer 2 Report

Dear authors and editor,

Congratulations on the research done. It is a very easy to read and simple research. That is to say, perfect. Science should be like that, simple, clear and transparent. 

I would like to congratulate you especially in the results section, because it is very well developed. The statistics is neat, you have followed all the steps, you have added the references of the instruments in each table. This part is great.

However, as a reviewer it is my duty to help you to get all the potential out of the research. 

In the methodology section we need some information.

In 2.2 participants: add the type of sampling, if it was a convenience sample, if it was by clusters, etc. I understand that being with students it was by convenience, since some centers would want to participate and others would not, and that you could not choose randomly. This is normal, and there is nothing wrong with stating that the sample is for convenience. Obviously, whether or not the centers want to participate is essential. Also, please explain a bit about how the process went, if you contacted another institution to mediate, if you did so with associate professors from your university who work in these centers, etc. Also, add that you requested informed consents from the families, etc. This whole process, which we always and all researchers carry out, should be made clear in this section.
Also, when you say that you have studied the "grade of the participants", are you referring to the academic year? Add this information together with the percentage of men, women and age and average age. Add how many centers are enrolled in average and low economic data.  And it would even be very interesting, if you have the average or low socio-economic data, to calculate if there are differences. Perform an Xi-square test and add in this section the results or at the beginning of the results. Also, add in table 1 the Xi 2 test, to see if there are differences or not between socio-cultural environments. This is something complementary that enriches the research. In this way we will be able to see if the interventions in your country are improving the situation or if more work needs to be done on them.
In section 2.3.1 add the questionnaire in your mother tongue as complementary material.

As for the figures, could you add some color to them? You can put them through an editing process or simply in power point. An aesthetic figure is important, it captures the attention and helps the reader. Remember to avoid greens and reds, use a blue tone instead.

The discussion section is very well developed, but as a limitation and future studies, it would be very interesting to mention behavioral addictions to social networks and the Internet. Add in line 132 something like: "Currently, young people and adolescents do not only present Substance Use. Addiction to social networks and the Internet is growing and is a new challenge [57,58,59]. New research is needed to study both substance and behavioral addictions, whether social networking or online gaming, to determine the moderating role they might present."

New references:
57. Lozano-Blasco, R.; Cortés-Pascual, A. Problematic internet uses and depression in adolescents: A meta-analysis. Comunicar 2020, 28, 109-120. 
58. Yan, W.; Li, Y.; Sui, N. The relationship between recent stressful life events, personality traits, perceived family functioning and internet addiction among college students. Stress Health 2014, 30, 3-11. [CrossRef]
59. Akbari, M.; Seydavi, M.; Spada, M.M.; Mohammadkhani, S.; Jamshidi, S.; Jamaloo, A.; Ayatmehr, F. The Big Five personality traits and online gaming: A systematic review and meta-analysis. J. Behav. Addict. 2021, 10, 611-625. 

Author Response

(The authors gave the same response as above.)

Reviewer 3 Report

The work entitled The Mediating Role of Contextual Problems and Sensation Seeking in the Association between Substance Use and Mental Health in Adolescents from Northern Chile” contains new scientific knowledge and covers a relevant topic. However, I have some comments that have to be addressed before it can be considered for publication.

The introduction contains relevant literature, however some of them are outdated. I suggest reviewers to introduce recent relevant citations related to both adolescents’ mental health and substance use. For example, and among others:  Caspi, Houts, Ambler et al., 2020; Montgomery, Vaughn, Jacquez (2022); Sánchez-García, Ortuño-Sierra, Paino, and Fonseca-Pedrero (2021).

With this regard, both in the introduction and in the discussion, authors may consider devoting some lines to possible educational implications of the study (see Montgomery et al., 2022)

In my opinion, the objectives of the study should be further developed. In addition, the hypotheses of the study should be introduced.

In the participants section, authors did not specify if they ask for previous history of mental problems. It is likely than in a sample over 2000 participants, some of them have mental problems that are affecting the interpretation of the results.

Also, the description of the participants is rather weak. Authors do not explain for instance age distribution or nationality. In addition, how was the sampling method (e.g. incidental, stratified, etc.). Also, how did authors manage possible outliers? Also, was there any exclusion criteria for participation in the study?

Author Response

(The authors gave the same response as above.)

Round 2

Reviewer 3 Report

Authors have addressed all my previous comments.

I have no further suggestions.